# A sepsis treatment algorithm to improve early antibiotic de-escalation while maintaining adequacy of coverage (Early-IDEAS): A prospective observational study

Mohamed Abdulla Ghuloom Abdulla Bucheeri[1]*, Marion Elligsen[2], Philip W. Lam[2,3], Nick Daneman[2,3], Derek MacFadden[4]

1 Department of Medicine, University of Toronto, Toronto, Ontario, Canada, 2 Sunnybrook Health Sciences Centre, Toronto, Ontario, Canada, 3 Division of Infectious Diseases, University of Toronto, Toronto, Canada, 4 The Ottawa Hospital Research Institute, Ottawa, Ontario, Canada

* mohamed.bucheeri@mail.utoronto.ca

## Abstract

### Background

Empiric antibiotic treatment selection should provide adequate coverage for potential pathogens while minimizing unnecessary broad-spectrum antibiotic use. We sought to pilot a sepsis treatment algorithm to individualize antibiotic recommendations, and thereby improve early antibiotic de-escalation while maintaining adequacy of coverage (Early-IDEAS).

### Methods

In this observational study, the Early-IDEAS decision support algorithm was derived from previous Gram-negative and Gram-positive prediction rules and models along with local guidelines, and then applied to prospectively identified consecutive adults within 24 hours of suspected sepsis. The primary outcome was the proportion of patients for whom de-escalation of the primary antibiotic regimen was recommended by the algorithm. Secondary outcomes included: (1) proportion of patients for whom escalation was recommended; (2) number of recommended de-escalation steps along a pre-specified antibiotic cascade; and (3) adequacy of therapy in patients with culture-confirmed infection.

### Results

We screened 578 patients, of whom 107 eligible patients were included. The Early-IDEAS treatment recommendation was informed by Gram-negative models in 76 (71%) patients, Gram-positive rules in 64 (59.8%), and local guidelines in 27 (25.2%). Antibiotic de-escalation was recommended in almost half of all patients (n = 52, 48.6%), with a median of 2 steps down the *a priori* antibiotic treatment cascade. No treatment change was recommended in 45 patients (42.1%), and escalation was recommended in 10 (9.3%). Among the 17 patients with positive blood cultures, both the clinician prescribed regimen and the algorithm recommendation provided adequate coverage for the isolated pathogen in 12 patients

**Data Availability Statement:** Clinical data is available as presented in this paper and supplements. Parties interested in patient-level

data can contact the authors for consideration of sharing via data transfer agreement in alignment with our institutional policies.

**Funding:** The authors received no specific funding for this work.

**Competing interests:** The authors have declared that no competing interests exist.

(70.6%), (p = 1). Among the 25 patients with positive relevant, non-blood cultures, both the clinician prescribed regimen and the algorithm recommendation provided adequate coverage in 20 (80%), (p = 1).

## Conclusion

An individualized decision support algorithm in early sepsis could lead to substantial antibiotic de-escalation without compromising adequate antibiotic coverage.

## Introduction

Antibiotic resistance is recognized as one of the greatest public health challenges of our time and threatens the sustained availability of effective treatments for common infectious diseases [1]. Antibiotic use has been shown to be the major driver of antibiotic resistance, and interventions to reduce unnecessary prescribing are urgently needed [2].

However, as antibiotic resistance increases globally, it becomes more difficult to select and provide adequate empiric antibiotic therapy while adhering with the principles of antibiotic stewardship, particularly in patients with life-threatening infections who stand to benefit the most from early adequate treatment [3, 4]. Suspected sepsis is one of the most common indications for the empiric use of broad spectrum antibiotics given the high short-term mortality risk associated with this condition and need to provide timely effective therapy [5]. However, the strategy of simply broadening the spectrum of empiric antibiotic treatment for all patients is not ideal, as many individuals will have infections that can be treated with narrower antibiotics, and this practice likely favors development of further resistance to reserve agents both for the individual and populations.

Current available evidence suggests that the use of computerized clinical decision support tools can increase the percentage of patients that receive desired care, but robust evidence on the use of this approach in patients with suspected sepsis is limited [6]. Two previous studies indicated that clinical decision support algorithms including mathematical models and rules based on an individual's historic microbiology results, could be used to predict individual risk factors for resistance and successfully guide antibiotic selection [7, 8]. While effective, these approaches were limited by their focus on either Gram-positive or Gram-negative pathogens (separately) and were applied only after preliminary culture results were available. To improve on these, we have developed an encompassing algorithm, using modelling and rules, that considers all potential bacteria and can be used at the most critical empiric window (prior to microbiologic results). This early sepsis treatment algorithm could permit a narrower spectrum empiric therapy, while supporting the adequacy of coverage, allowing each patient to be on the right drug at the right time. This approach offers the potential for maximal benefits in adequate coverage and antibiotic stewardship by intervening earlier at the time of clinical presentation. However, early intervention poses additional challenges, and so our approach requires further validation before large-scale clinical implementation.

In this observational study, adult patients admitted at a large academic tertiary care center were prospectively reviewed to evaluate an early sepsis treatment algorithm. We hypothesized that the use of this new algorithm would have the potential to support early de-escalation recommendations while maintaining or improving adequacy of treatment.

## Methods

### Study setting, design, and participants

We performed a prospective observational study to evaluate the expected impact of an early sepsis treatment algorithm. This study was carried out at Sunnybrook Health Sciences Centre in Ontario, Canada, from November to December 2021. The ethics committee of Sunnybrook Research Institute has waived ethics approval and individual consent requirement for this project as it was deemed a quality improvement project by Sunnybrook Research Institute Ethics Review–Self Assessment Tool (ER–SAT). Clinical data is available as presented in this paper and supplements. Parties interested in patient-level data can contact the authors for consideration of sharing via data transfer agreement in alignment with our institutional policies.

### Patient inclusion and exclusion criteria

The following inclusion criteria were used to identify adults with suspected early sepsis requiring empiric antibiotic treatment: (1) adult aged 18 years or older; (2) admitted to hospital; (3) received an eligible systemic antibiotic (See Fig 1); and (4) had blood cultures ordered within ±12 hours of receipt of antibiotics. The following exclusion criteria were applied: (1) patients already included in the study during prior episodes of early sepsis; (2) those receiving palliative care; (3) those who were pregnant; (4) other inpatient antibiotic use in the prior 72 hours; and (5) positive clinical culture associated within the index event available at the time of assessment. A roster of patients was generated twice daily during business hours Monday—Friday to identify patients that met the above criteria within the prior 24 hours. Patient charts were reviewed in real-time, and the early sepsis treatment algorithm was applied to determine the recommended antibiotic regimen for eligible patients.

### Early sepsis treatment algorithm

We evaluated an early sepsis decision algorithm (Fig 1), which provided the empiric antibiotic selection in the context of suspected infection. This combined sepsis treatment algorithm was derived from rules and modelled prediction approaches that were developed and tested for both Gram-positive and Gram-negative infections [7, 8]. The Gram-negative prediction model is as previously described [7], and in brief consists of multiple parametric regression models which predict the likelihood of susceptibility for each commonly used antibiotic for Gram-negative pathogens, based on epidemiologic predictors (age, sex, prior hospitalization, prior ICU stay, prior antibiotic exposure) and prior culture results (prior antibiotic-resistant organisms from clinical cultures in the preceding year). These Gram- negative parametric models were validated and calibrated on historical culture data from the institution under study [7]. The Gram-positive algorithm is based on a previous prospective observational study and directs the addition or cessation of vancomycin based on prior methicillin-resistant *Staph. aureus* (MRSA)screen cultures (Fig 1) [8]. For infections treated with a standard empiric regimen, regardless of patient 's risk factors, local guidelines for empiric therapy were applied (this included cellulitis, community- acquired pneumonia and meningitis). The treatment algorithm assumes an *a priori* antibiotic cascade for the treatment of Gram-negative pathogens, with the following antibiotics considered from broadest to narrowest respectively (meropenem>piperacillin- tazobactam>ceftazidime>ceftriaxone>ciprofloxacin). It seeks to move the prescriber down the antibiotic selection cascade to generally narrower spectrum agents by recommending the narrowest spectrum agent that still exceeds a pre-specified threshold of adequate coverage (80% for patients with a quick sequential organ failure assessment (qSOFA) score <2, 90% for patients with a qSOFA score > 2 or receiving vasopressor support) [9].

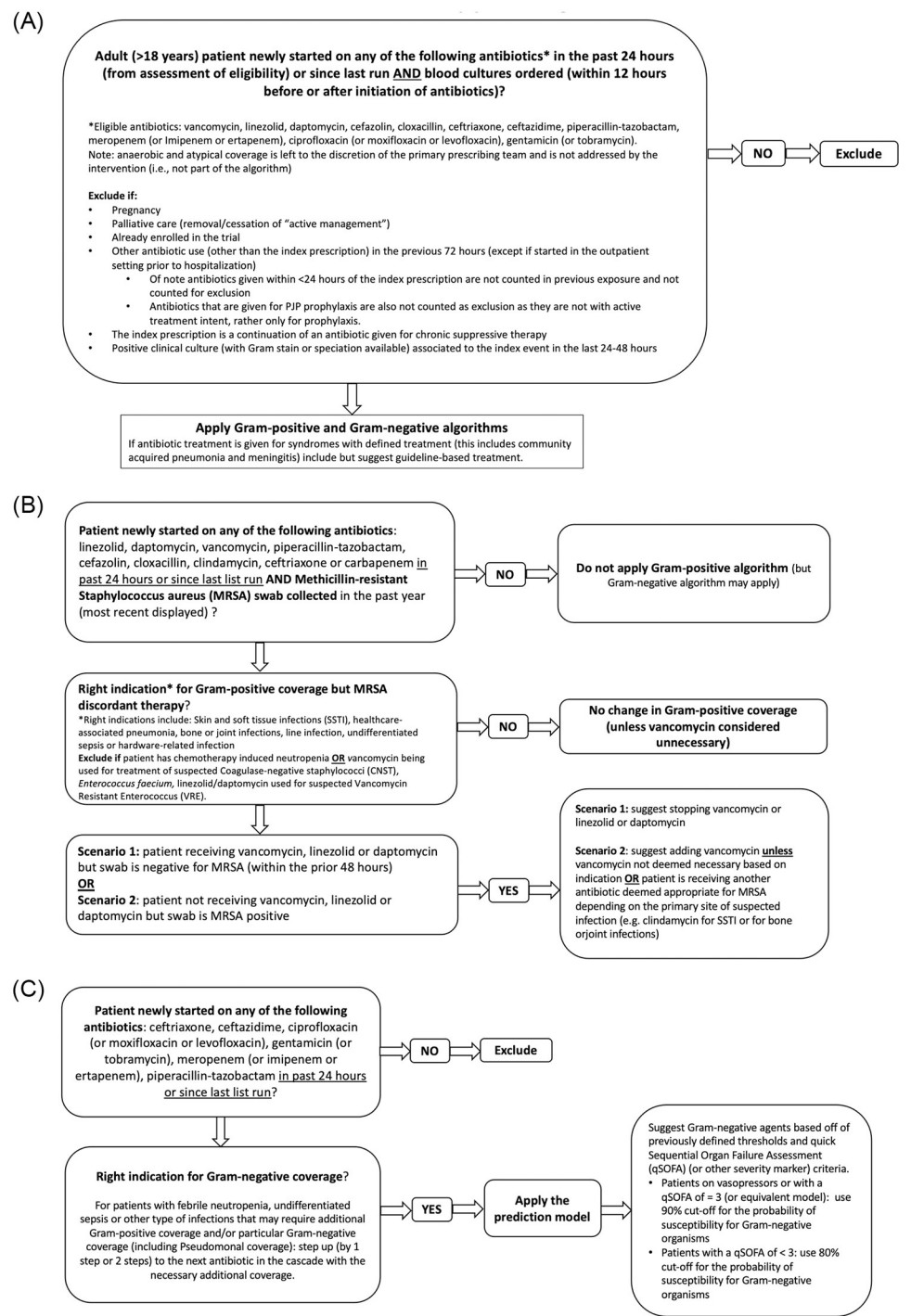

**Fig 1.** A. Eligibility for the algorithm. B. Gram-positive coverage component of the Early-IDEAS Algorithm. C. Gram-negative coverage component of the Early-IDEAS Algorithm.

When the clinical syndrome was deemed to require Gram-positive coverage (other than MRSA coverage) which the narrowest choice did not provide, the next antibiotic that provided Gram-positive coverage, moving up the cascade, was selected as the appropriate antibiotic to cover the clinical syndrome. The purpose of this pilot study was to evaluate the potential

impact of an early sepsis treatment algorithm, and therefore we did not provide treatment recommendations to the clinical team; the results of this pilot will inform further rigorous prospective evaluation.

## Outcomes and predictor variables

The main outcome measure of this study was the proportion of patients whom de-escalation from the primary antibiotic regimen was recommended by our proposed approach. De-escalation was defined as movement down the Gram-negative antibiotic cascade or the cessation of vancomycin or daptomycin. Escalation was defined as movement up the Gram-negative antibiotic cascade or addition of vancomycin or daptomycin. If antibiotics were not changed, then these were considered as 'No change' in spectrum. In some instances, antibiotics were changed to those not included in our antibiotic cascade (e.g., levofloxacin for community acquired pneumonia by local guidelines), or atypical antimicrobial coverage was added (e.g., azithromycin for atypical organism coverage in CAP). For these instances we typically considered these guideline-based changes as equivalent in spectrum (e.g., we classified guideline directed prescribing with levofloxacin or ceftriaxone/azithromycin for CAP as equivalent spectrum-level). We also assumed no change in spectrum-level with the addition of atypical or anaerobic specific adjunctive agents such as azithromycin or metronidazole. We considered the following agents not listed in our antibiotic cascade as equivalent levels of spectrum (i.e., there would be no change in spectrum with a change of agent): (1) ertapenem and meropenem; and (2) ceftazidime and gentamicin.

Secondary outcomes that we evaluated included: (1) the proportion of patients where escalation was recommended; (2) the number of de-escalation steps along the antibiotic cascade that would be achieved; and (3) the proportions of culture-positive patients who would receive adequate therapy with a given regimen (suggested vs. actual). Adequacy of therapy could only be determined for the subset of patients with a culture-positive infection. To describe the study population and stratify by relevant factors, we collected the following predictor variables: age, sex, qSOFA score [10], neutropenia, vasopressor use, mechanical ventilation, antibiotics prescribed at the time of assessment, clinical cultures, and prior MRSA screening cultures.

## Statistical analysis

The primary and secondary outcomes were described using proportions for binary and categorical variables and medians and interquartile ranges for continuous variables. A comparison between the proportion of patients receiving adequate therapy for suggested versus received antibiotic therapy was performed using Fisher's exact test. Some outcomes were stratified by relevant covariates.

## Results

### Patient characteristics

We reviewed 578 charts for eligibility during the study period, of which 471 were excluded (Fig 2). The majority of included patients received an antibiotic that had some degree of Gram-negative activity (92.5%), whereas only 8 patients received only Gram-positive active agents. The characteristics of the included patients are shown in Table 1. Less than half of the patients were female (n = 42, 39.3%), the mean age was 66.3 years, and the majority (n = 80, 74.8%) of patients had suspected community-acquired infection. Most patients were not markedly ill (median qSOFA score of 1), had undifferentiated sepsis (n = 26, 24.3%), and were most frequently prescribed piperacillin-tazobactam (n = 52, 48.6%) for empiric therapy. These

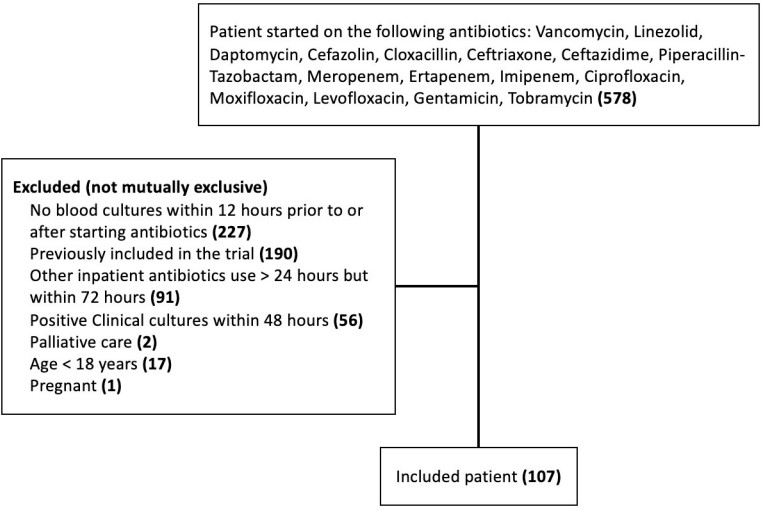

**Fig 2. Study flow diagram.**

characteristics were relatively consistent across the patients who were de-escalated, escalated, or had no change to therapy.

Our treatment algorithm was applied to all eligible patients, and treatment recommendation was informed by the Gram-negative models in 76 patients (71%), by local guidelines in 27 (25.2%) and by Gram-positive rules in 64 (59.8%).

## Treatment recommendations

Antibiotic de-escalation was recommended by the algorithm in almost half of all patients (n = 52, 48.6%), no treatment change was recommended in 45 patients (42.1%), and escalation was recommended in 10 patients (9.3%) (Tables 2 and 3). Amongst the patients where de-escalation was recommended, the median number of steps down the *a priori* antibiotic treatment cascade was 2 (Table 3). Table 2 displays the antibiotics administered to patients, categorized by whether they were recommended by the Early-IDEAS algorithm for escalation, de-escalation, and no change, as well as the corresponding percentage distribution. This data is presented for both the antibiotics that were being prescribed by the medical team and those that were recommended by the algorithm. Many of the recommended de-escalation steps according to the Early-IDEAS algorithm involved the de-escalation of piperacillin-tazobactam to a narrower spectrum agent in the antibiotic cascade. In the small number of patients who were recommended escalation by algorithm, half of these (5/10, 50%) involved changing patients' antimicrobial recommendation from piperacillin-tazobactam to meropenem due to a significant predicted risk for resistant Gram-negative pathogens.

## Adequacy of coverage

A substantial subset of patients (n = 37, 34.6%) were ultimately confirmed to have culture-positive infection whereby susceptibility results could be used to determine the adequacy of recommended coverage. For 17 patients with clinically relevant positive blood cultures, adequate antibiotics were prescribed by the clinical team in 12 (70.6%) patients. When applying the treatment algorithm for these same patients, the percentage of adequate empiric antibiotic

**Table 1. Characteristics of patients overall and among those for whom the Early-IDEAS treatment algorithm recommended de-escalation, escalation, or no change in therapy.** SD = Standard Deviation, IQR = Interquartile range, ICU = Intensive Care Unit, qSOFA = quick sequential organ failure assessment, CNS = Central Nervous System, UTI = Urinary Tract Infection, MRSA = Methicillin-Resistant *Staphylococcus aureus.*

| Patient Characteristic | All Patients (n = 107) | De-escalation (n = 52) | Escalation (n = 10) | No Change (n = 45) |
|---|---|---|---|---|
| Male Sex | 65 (60.7%) | 30 (57.7%) | 6 (60%) | 29 (64.4%) |
| Age, mean (SD) | 66.3 (19.7) | 67.4 (18.9) | 66.8 (18.6) | 64.8 (21.2) |
| Prior Hospital stay in 90 days | 34 (31.8%) | 18 (34.6%) | 7 (70%) | 9 (20%) |
| Hospital acquired | 27 (25.2%) | 9 (17.3%) | 5 (50%) | 13 (28.9%) |
| Prior ICU stay in 90 days | 9 (8.4%) | 3 (5.8%) | 5 (50%) | 1 (2.2%) |
| ICU acquired | 6 (5.6%) | 1 (1.9%) | 0 (0%) | 5 (11.1%) |
| Surgical Service admission | 25 (23.4%) | 11 (21.2%) | 4 (40%) | 10 (22.2%) |
| qSOFA Score | | | | |
| 0 | 50 (46.7%) | 24 (46.2%) | 6 (60%) | 20 (44.4%) |
| 1 | 36 (33.6%) | 16 (30.8%) | 2 (20%) | 18 (40%) |
| 2 | 16 (15%) | 10 (19.2%) | 1 (10%) | 5 (11.1%) |
| 3 | 5 (4.7%) | 2 (3.8%) | 1 (10%) | 2 (4.4%) |
| Median (IQR) | 1 (0–1) | 1 (0–1) | 0 (0–1) | 1 (0–1) |
| On Vasopressors | 10 (9.3%) | 7 (13.5%) | 1 (10%) | 2 (4.4%) |
| On Mechanical Ventilation | 6 (5.6%) | 2 (3.8%) | 1 (10%) | 3 (6.7%) |
| Infection Syndromes | | | | |
| CNS infection | 2 (1.9%) | 0 (0%) | 0 (0%) | 2 (4.4%) |
| Febrile neutropenia | 13 (12.1%) | 2 (3.8%) | 3 (30%) | 8 (17.8%) |
| Hepatobiliary | 4 (3.7%) | 2 (3.8%) | 2 (20%) | 0 (0%) |
| Intra-abdominal | 7 (6.5%) | 5 (9.6%) | 1 (10%) | 1 (2.2%) |
| Odontogenic infection | 2 (1.9%) | 1 (1.9%) | 0 (0%) | 1 (2.2%) |
| Pneumonia | 24 (22.4%) | 10 (19.2%) | 0 (0%) | 14 (31.1%) |
| Skin and soft tissue / bone | 15 (14%) | 6 (11.5%) | 1 (10%) | 8 (17.8%) |
| Unknown/Undifferentiated | 26 (24.3%) | 14 (26.9%) | 3 (30%) | 9 (20%) |
| UTI | 14 (13.1%) | 12 (23.1%) | 0 (0%) | 2 (4.4%) |
| Clinical Cultures | | | | |
| Any positive culture | 47 (43.9%) | 21 (40.4%) | 6 (60%) | 20 (44.4%) |
| Clinically Relevant cultures | 37 (78.7%) | 16 (43.2%) | 5 (13.5%) | 16 (43.2%) |
| Blood cultures | 23 (21.5%) | 10 (19.2%) | 4 (40%) | 9 (20%) |
| Clinically relevant | 17 (73.9%) | 10 (100%) | 3 (75%) | 4 (44.4%) |
| Positive cultures excluding blood cultures | 31 (29%) | 14 (26.9%) | 2 (20%) | 15 (33.3%) |
| Clinically relevant | 25 (80.6%) | 9 (64.3%) | 2 (100%) | 14 (93.3%) |
| MRSA Swabs done | 64 (59.8%) | 33 (63.5%) | 10 (100%) | 21 (46.7%) |
| Positivity rate | 1 (1.6%) | 1 (3%) | 0 (0%) | 0 (0%) |

recommendations was unchanged (70.6%, p = 1), and 7 of these 12 patients would have been de-escalated from their given therapy. For 25 patients with clinically relevant positive (non-blood) clinical cultures, adequate antibiotics were prescribed in 20 (80%) patients by the clinical team. When applying the treatment algorithm for these same patients, the percentage of adequate treatment recommendations was unchanged (80%, p = 1), and 6 of these 20 patients would have been de-escalated from their given therapy. For 5 patients with clinically relevant cultures in both blood and non-sterile sites, adequate antibiotics were prescribed in 3 (60%) patients by the clinical team. When applying the treatment algorithm for these same patients, the percentage of treatment recommendations was unchanged (60%, p = 1), and 2 of these 5 patients had adequate coverage through de-escalation.

**Table 2. Distribution of antibiotics received by the patients compared with those antibiotics recommended by the Early-IDEAS algorithm.** These are further stratified by whether the Early-IDEAS algorithm had recommended escalation, de-escalation, or no change. IQR = Interquartile range.

| | Antibiotics Received (Prescribed by Team) | | | | Antibiotics Recommended by Early-IDEAS Algorithm | | | |
|---|---|---|---|---|---|---|---|---|
| | All Patients (n = 107) | De-escalation (n = 52) | Escalation (n = 10) | No Change (n = 45) | All Patients (n = 107) | De-escalation (n = 52) | Escalation (n = 10) | No Change (n = 45) |
| Ampicillin | 1 (0.9%) | 0 (0%) | 0 (0%) | 1 (2.2%) | 1 (0.9%) | 0 (0%) | 0 (0%) | 1 (2.2%) |
| Azithromycin | 9 (8.4%) | 3 (5.8%) | 0 (0%) | 6 (13.3%) | 24 (22.4%) | 11 (21.2%) | 0 (0%) | 13 (28.9%) |
| Ciprofloxacin | 1 (0.9%) | 0 (0%) | 1 (10%) | 0 (0%) | 18 (16.8%) | 18 (34.6%) | 0 (0%) | 0 (0%) |
| Cefazolin | 10 (9.3%) | 0 (0%) | 3 (30%) | 7 (15.6%) | 8 (7.5%) | 1 (1.9%) | 0 (0%) | 7 (15.6%) |
| Ceftazidime | 1 (0.9%) | 1 (1.9%) | 0 (0%) | 0 (0%) | 1 (0.9%) | 0 (0%) | 0 (0%) | 1 (2.2%) |
| Ceftriaxone | 34 (31.8%) | 12 (23.1%) | 1 (10%) | 21 (46.7%) | 55 (51.4%) | 31 (59.6%) | 2 (20%) | 22 (48.9%) |
| Cephalexin | 1 (0.9%) | 0 (0%) | 1 (10%) | 0 (0%) | 0 (0%) | 0 (0%) | 0 (0%) | 0 (0%) |
| Clindamycin | 1 (0.9%) | 1 (1.9%) | 0 (0%) | 0 (0%) | 0 (0%) | 0 (0%) | 0 (0%) | 0 (0%) |
| Doxycycline | 2 (1.9%) | 1 (1.9%) | 0 (0%) | 1 (2.2%) | 0 (0%) | 0 (0%) | 0 (0%) | 0 (0%) |
| Ertapenem | 2 (1.9%) | 1 (1.9%) | 0 (0%) | 1 (2.2%) | 0 (0%) | 0 (0%) | 0 (0%) | 0 (0%) |
| Gentamicin | 1 (0.9%) | 0 (0%) | 0 (0%) | 1 (2.2%) | 0 (0%) | 0 (0%) | 0 (0%) | 0 (0%) |
| Levofloxacin | 1 (0.9%) | 0 (0%) | 0 (0%) | 1 (2.2%) | 0 (0%) | 0 (0%) | 0 (0%) | 0 (0%) |
| Metronidazole | 9 (8.4%) | 4 (7.7%) | 2 (20%) | 3 (6.7%) | 13 (12.1%) | 9 (17.3%) | 1 (10%) | 3 (6.7%) |
| Meropenem | 5 (4.7%) | 4 (7.7%) | 0 (0%) | 1 (2.2%) | 8 (7.5%) | 0 (0%) | 6 (60%) | 2 (4.4%) |
| Piperacillin-Tazobactam | 52 (48.6%) | 34 (65.4%) | 5 (50%) | 13 (28.9%) | 17 (15.9%) | 2 (3.8%) | 2 (20%) | 13 (28.9%) |
| Vancomycin | 11 (10.3%) | 5 (9.6%) | 0 (0%) | 6 (13.3%) | 8 (7.5%) | 2 (3.8%) | 0 (0%) | 6 (13.3%) |

## Discussion

In this prospective validation of the study of the Early-IDEAS decision support algorithm we show that patients with suspected sepsis can be treated with narrower antibiotic agents while maintaining the adequacy of initial empiric coverage. The decision support algorithm we employed recommended de-escalation in almost half of all eligible patients, with a high proportion of blood and non-blood isolates adequately covered. Less than 1 in 10 patients had an escalation of antibiotics recommended. The Gram-negative prediction used, in brief, consists of multiple parametric regression models. The treatment algorithm assumes an *a priori*

**Table 3. (A) Summary of antibiotic step changes by those with Early-IDEAS algorithm recommendation of de-escalation or escalation; (B) Summary of Early-IDEAS algorithm recommendations, by aspects of algorithm utilized, and with expected results (de-escalation, escalation, or no change).**

**A.**

| Antibiotic step change | De-escalation | Escalation |
|---|---|---|
| 1 | 15 (28.8%) | 6 (60%) |
| 2 | 29 (55.8%) | 2 (20%) |
| 3 | 7 (13.5%) | 1 (10%) |
| 4 | 1 (1.9%) | 0 (0%) |
| 5 | 0 (0%) | 1 (10%) |
| Median (IQR) | 2(1–2) | 1(1–2) |

**B.**

| | All Patients (n = 107) | De-escalation (n = 52) | Escalation (n = 10) | No Change (n = 45) |
|---|---|---|---|---|
| Guideline applied | 27 (25.2%) | 8 (15.4%) | 0 (0%) | 19 (42.2%) |
| Gram-negative Algorithm applied | 76 (71%) | 44 (84.6%) | 8 (80%) | 24 (53.3%) |
| Gram-positive model applied | 64 (59.8%) | 33 (63.5%) | 10 (100%) | 21 (46.7%) |

antibiotic cascade for the treatment of Gram-negative pathogens and seeks to move the prescriber down the antibiotic selection cascade to generally narrower spectrum agents.

## Interpretation

Clinical decision support tools for selecting antimicrobials in infectious syndromes are not new [11]; however, studies evaluating sepsis-specific support are less frequent. To date, there have been a few studies evaluating support tools in the management of sepsis, though they are often focused on other (non-prescribing) aspects of management, including identifying prognosis and severity, determining likely discharge disposition, and the need for early treatment [12–14]. We did not identify any decision support tools developed for empiric antibiotic selection in early sepsis that have been evaluated in a prospective randomized fashion, and there is a clear need for validated support models that can be tested in rigorous prospective trials.

Clinical decision support models, such as the Early-IDEAS treatment algorithm validated in this study, can be powered by institution-specific predictive models that could address the two competing aspects of antimicrobial stewardship: (1) the desire to retain high adequacy of empiric coverage while (2) reducing the breadth of antibiotic spectrum used to reduce the selection of resistance to reserve antimicrobial agents. Our algorithm demonstrated a significant number of patients can be de-escalated while maintaining a similar and adequate level of antibiotic coverage. Our approach fundamentally allows the provider to differentiate between patients that do not require broad-spectrum antibiotic therapy and those that do. Explicit, informed, and reproducible models have the potential to broadly support providers of all experience levels and backgrounds to operationalize antibiotic decision-making.

## Limitations & generalizability

This study is intended as a pilot of early clinical decision support, and so is not powered to evaluate downstream clinical benefits of early de-escalation (such as reduction in antimicrobial costs, complications, and resistance) or the downstream benefits of adequate coverage (such as rapidity of clinical cure, reduced lengths of hospital stay and increased survival). The mathematical algorithms underpinning the decision support are derived from culture-positive infections, and we cannot be certain that culture-negative infections are caused by the same distribution of pathogens; this limitation is also intrinsic to any antibiogram-based predictions which are used as a gold standard for supporting empiric prescribing [15].

Similarly, as anticipated, only a subset of enrolled patients (34.6%) had microbiologically confirmed and clinically relevant pathogens, and so for the remainder we are unable to compare the adequacy of coverage of the regimens recommended by the treating team versus our decision support algorithm. This is an expected trade-off of intervening early among patients presenting with sepsis. It is also important to note that the antibiotics evaluated in this study, and recommended by the Early-IDEAS algorithm, are systemic antibiotics which would penetrate the majority of infection sites.

This pilot was designed to refine our algorithms and did not involve actual communication of treatment recommendations to the treating team, so we are not yet able to confirm whether clinicians will accept this decision support. However, prior work has suggested high uptake of recommendations [7, 8], and so we expect high uptake with future implementation of this decision support. Lastly, we found that concurrent MRSA swab results were generally not available at the early time points of assessment for patients during this pandemic period, and thus the Gram-positive arm of the algorithm was of limited utility for recommending cessation of vancomycin.

## Conclusion

In summary, we demonstrate that a combined Gram-positive and Gram-negative decision support algorithm in early sepsis (Early-IDEAS) could help improve empiric antibiotic treatment by offering providers with narrower spectrum treatment options while maintaining high adequacy of therapy. This treatment algorithm requires further prospective evaluation to determine acceptability and efficacy. The time has come for the adoption of personalized medicine in sepsis treatment, and individualized models to support treatment selection may help us choose the right antibiotic at the right time.

## Author Contributions

**Conceptualization:** Mohamed Abdulla Ghuloom Abdulla Bucheeri, Marion Elligsen, Nick Daneman, Derek MacFadden.

**Data curation:** Mohamed Abdulla Ghuloom Abdulla Bucheeri.

**Formal analysis:** Mohamed Abdulla Ghuloom Abdulla Bucheeri, Nick Daneman, Derek MacFadden.

**Investigation:** Mohamed Abdulla Ghuloom Abdulla Bucheeri, Nick Daneman, Derek MacFadden.

**Methodology:** Nick Daneman, Derek MacFadden.

**Project administration:** Nick Daneman, Derek MacFadden.

**Resources:** Nick Daneman, Derek MacFadden.

**Supervision:** Marion Elligsen, Philip W. Lam, Nick Daneman, Derek MacFadden.

**Writing – original draft:** Mohamed Abdulla Ghuloom Abdulla Bucheeri, Marion Elligsen, Philip W. Lam, Nick Daneman, Derek MacFadden.

**Writing – review & editing:** Mohamed Abdulla Ghuloom Abdulla Bucheeri, Marion Elligsen, Philip W. Lam, Nick Daneman, Derek MacFadden.

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
