## [Editor Report · Decision Letter 0]

17 Feb 2023

PONE-D-23-03428A sepsis treatment algorithm to improve early antibiotic de-escalation while maintaining adequacy of coverage (Early-IDEAS): A prospective observational studyPLOS ONE

Dear Dr. BUCHEERI,

Thank you for submitting your manuscript to PLOS ONE. After careful consideration, we feel that it has merit but does not fully meet PLOS ONE’s publication criteria as it currently stands. Therefore, we invite you to submit a revised version of the manuscript that addresses the points raised during the review process.

 ========================The present decision is based on my inspection of the manuscript without input from external peer reviewers yet. This is because there are several major issues with the results which require clarification before the manuscript could be sent out for review. These are:1. Table 2 - the current presentation is confusing; the text (L253-261) accompanying the table described the recommendations in terms of patients, but the table presented the recommendations in relation to each antibiotic. Furthermore, it's unclear how the table is divided into 2 halves, antibiotics prescribed by team and recommended by algorithm. For example, for "no change" in ceftriaxone and meropenem, the numbers are 20 and 1 in the first half, but 22 and 2 in the second half.2. Table 3 - similar issue as table 2. It's unclear how the numbers for each recommendation category should be interpreted with respect to each column. Also, for this patient subgroup (n=37), it can be inferred that 5 have both positive blood and non-sterile site cultures positive results, but the antibiotics given would obviously be systemic and apply to both (in most cases, though there are exceptions like daptomycin being useful for bacteremia but not pneumonia). It would be important to verify that the antibiotics given or recommended be adequate to cover both categories of positive cultures.3. While keeping in mind that this is a pilot study, there are some notable results from table 3, (a) no difference in treatment adequacy between clinical team's prescription and algorithmic recommendations, and (b) there were more escalations than "no change" and de-escalations combined. When considered together, it appears reasonable to infer that the recommended escalations were unnecessary and/or inappropriate. Of course, this might be a misinterpretation due to issues with data presentation as mentioned above. Nonetheless, a more analytical discussion of the results in this subgroup would be necessary in the revision.========================

We look forward to receiving your revised manuscript.

Kind regards,

Herman Tse

Academic Editor

PLOS ONE
---

## [Author Response · Author response to Decision Letter 0]

11 Jun 2023

This is the revised manuscript after making edits as requested by the Academic editor (Dr. Rose Ann Joyce Sagun Puetes). Our response is as follows: 

"Thank you for your consideration of our manuscript "A sepsis treatment algorithm to improve early antibiotic de-escalation while maintaining adequacy of coverage (Early-IDEAS): A prospective observational study”, for publication in PLOS ONE. We just wanted to bring to your attention that the following remark:” In the Methods section please revise the informed consent statement to reflect whether written/verbal informed consent was obtained from all participants for inclusion in the study”, has been addressed as per the comments received from Dr. Paula Katrina A. Maderazo on April 29th, 2023. Our submission on May 8th, in (L183-185, L188) on the “Revised manuscript with track changes” document, stated “The ethics committee of Sunnybrook Research Institute has waived ethics approval and individual consent requirement for this project as it was deemed a quality improvement project by Sunnybrook Research Institute Ethics Review – Self Assessment Tool (ER–SAT)."

---

## [Decision Letter · Decision Letter 1]

4 Dec 2023

A sepsis treatment algorithm to improve early antibiotic de-escalation while maintaining adequacy of coverage (Early-IDEAS): A prospective observational study

PONE-D-23-03428R1

Dear Dr. BUCHEERI,

We’re pleased to inform you that your manuscript has been judged scientifically suitable for publication and will be formally accepted for publication once it meets all outstanding technical requirements.

Kind regards,

Herman Tse

Academic Editor

PLOS ONE

Additional Editor Comments (optional):

Reviewers' comments:

Reviewer's Responses to Questions

**Comments to the Author**

1. If the authors have adequately addressed your comments raised in a previous round of review and you feel that this manuscript is now acceptable for publication, you may indicate that here to bypass the “Comments to the Author” section, enter your conflict of interest statement in the “Confidential to Editor” section, and submit your "Accept" recommendation.

Reviewer #1: (No Response)

2. Is the manuscript technically sound, and do the data support the conclusions?

Reviewer #1: Yes

3. Has the statistical analysis been performed appropriately and rigorously? 

Reviewer #1: Yes

4. Have the authors made all data underlying the findings in their manuscript fully available?

Reviewer #1: Yes

5. Is the manuscript presented in an intelligible fashion and written in standard English?

Reviewer #1: Yes

6. Review Comments to the Author

Reviewer #1: The current research implementation is urgently required to enhance the efficacy and adequacy of personalized empirical antibiotic treatment in septic patients. The concerns that have emerged are clearly outlined within the limitations of the manuscript.

7. PLOS authors have the option to publish the peer review history of their article (what does this mean?). If published, this will include your full peer review and any attached files.

Reviewer #1: **Yes: **Sairah Hafeez Kamran

---

## [Editor Report · Acceptance letter]

10 Dec 2023

PONE-D-23-03428R1 

A sepsis treatment algorithm to improve early antibiotic de-escalation while maintaining adequacy of coverage (Early-IDEAS): A prospective observational study 

Dear Dr. Bucheeri:

I'm pleased to inform you that your manuscript has been deemed suitable for publication in PLOS ONE. Congratulations! Your manuscript is now with our production department. 

Kind regards, 

on behalf of

Dr. Herman Tse 

Academic Editor

PLOS ONE